# Supervised Segmentation of Ultra-High-Density Drone Lidar for Large-Area Mapping of Individual Trees

**Martin Krůček [1,2,\*]**, **Kamil Král [1]**, **KC Cushman [3]**, **Azim Missarov [1,2]** and **James R. Kellner [3,4]**

1 Department of Forest Ecology, The Silva Tarouca Research Institute, 60200 Brno, Czech Republic; kamil.kral@vukoz.cz (K.K.); azim.missarov@gmail.com (A.M.)
2 Faculty of Forestry and Wood Technology, Mendel University in Brno, 61300 Brno, Czech Republic
3 Department of Ecology and Evolutionary Biology, Brown University, Providence, RI 02912, USA; katherine_cushman@brown.edu (K.C.); james_r_kellner@brown.edu (J.R.K.)
4 Institute at Brown for Environment and Society, Brown University, Providence, RI 02912, USA
\* Correspondence: krucek.martin@gmail.com

**Abstract:** We applied a supervised individual-tree segmentation algorithm to ultra-high-density drone lidar in a temperate mountain forest in the southern Czech Republic. We compared the number of trees correctly segmented, stem diameter at breast height (DBH), and tree height from drone-lidar segmentations to field-inventory measurements and segmentations from terrestrial laser scanning (TLS) data acquired within two days of the drone-lidar acquisition. Our analysis detected 51% of the stems >15 cm DBH, and 87% of stems >50 cm DBH. Errors of omission were much more common for smaller trees than for larger ones, and were caused by removal of points prior to segmentation using a low-intensity and morphological filter. Analysis of segmented trees indicates a strong linear relationship between DBH from drone-lidar segmentations and TLS data. The slope of this relationship is 0.93, the intercept is 4.28 cm, and the $r^2$ is 0.98. However, drone lidar and TLS segmentations overestimated DBH for the smallest trees and underestimated DBH for the largest trees in comparison to field data. We evaluate the impact of random error in point locations and variation in footprint size, and demonstrate that random error in point locations is likely to cause an overestimation bias for small-DBH trees. A Random Forest classifier correctly identified broadleaf and needleleaf trees using stem and crown geometric properties with overall accuracy of 85.9%. We used these classifications and DBH estimates from drone-lidar segmentations to apply allometric scaling equations to segmented individual trees. The stand-level aboveground biomass (AGB) estimate using these data is 76% of the value obtained using a traditional field inventory. We demonstrate that 71% of the omitted AGB is due to segmentation errors of omission, and the remaining 29% is due to DBH estimation errors. Our analysis indicates that high-density measurements from low-altitude drone flight can produce DBH estimates for individual trees that are comparable to TLS. These data can be collected rapidly throughout areas large enough to produce landscape-scale estimates. With additional refinement, these estimates could augment or replace manual field inventories, and could support the calibration and validation of current and forthcoming space missions.

**Keywords:** aboveground biomass; carbon; laser scanning; object-based; population; remote sensing

---

## 1. Introduction

Numerous studies have demonstrated the utility of terrestrial laser scanning (TLS) for automated segmentation of individual trees in high-density point clouds [1–3]. Because these data can be acquired and processed rapidly in comparison to the time required for traditional field inventories, they have

the potential to increase the accuracy and frequency of stand-level forest assessment. However, scaling these methods to landscapes larger than a few hectares has been challenging [1,2].

Drone remote sensing may be able to overcome this challenge [4,5]. Lidar sensors on low-altitude drones can produce measurement densities in the thousands of points per square meter from wide scan angles that clearly resolve individual stem and branch structure [6]. Previous work has demonstrated that tree stems can be manually extracted from a high-resolution point cloud acquired by a low-altitude drone in a European temperate broadleaf forest [7,8]. Diameter at breast height (DBH) estimates using these data are strongly correlated with DBH of the same trees quantified using TLS [7], and a study in eucalypt forest in Australia showed that tree DBH from TLS data are unbiased with respect to field measurements [6]. In fact, because high-density lidar data can model wood volume by segmenting individual trees [3,9], lidar data provide better estimates of tree-level aboveground biomass (AGB) and wood volume than manual methods using diameter at breast height (DBH) and allometric scaling equations [6,10].

However, data from drone lidar differ from TLS measurements in important ways that create challenges to individual tree segmentation [5]. In general terms, data from drone lidar are about one order of magnitude less dense than TLS measurements, contain about one order of magnitude more noise from random and systematic components, and are acquired using footprints that are about one order of magnitude larger than TLS data. A recent analysis showed that manually identified individual trees could be segmented and their AGB quantified using drone lidar in a temperate broadleaf forest [4]. It remains unclear whether uncertainties in drone lidar data undermine our ability to apply individual tree segmentation to data sets large enough to support the calibration and validation activities of current and forthcoming space missions [11], including the NASA Global Ecosystem Dynamics Investigation [12], ESA BIOMASS [13], and the NASA-ISRO Synthetic Aperture Radar (NISAR; [14]).

The challenge is consistent application of segmentation algorithms to large data sets. Individual tree segmentation algorithms developed for TLS data require optimization to local structural conditions [9,15]. Because applications to date have been applied to relatively small areas up to a few hectares in size, it is not clear whether these algorithms can be generally applied to forests with varying structural conditions, or if they need to be adjusted to deal with local environments. Moreover, automated tree segmentation algorithms for airborne lidar data are usually based on top-down approaches, such as watershed segmentation [16,17]. Here we test bottom-up segmentation typically applied to TLS data that has greater ability to deal with complex stand structure. This test is facilitated by recent advances in ultra-high-density drone lidar technology [5].

We apply a new individual tree segmentation algorithm called 3D Forest [15] to ultra-high-density drone lidar and coincident TLS measurements in an old-growth temperate mountain forest in the south Bohemia region of the Czech Republic. The data were acquired using the Brown Platform for Autonomous Remote Sensing (BPAR; [5]). By generating a mean measurement density that exceeds 4000 points per square meter from scan angles out to 60 degrees, these data provide observations of stem and branch structure that are suitable for individual tree segmentation [5]. We compared stem diameter and height from automatically segmented tree objects to field measurements and data extracted from TLS scans that we acquired within two days of drone lidar acquisition. Here we use these data to ask what fraction of trees can be automatically segmented and measured using drone lidar in a 25 ha permanent inventory plot, and we develop a Random Forest classifier to identify tree segments as broadleaf or needleleaf trees. We then apply allometric scaling equations to all segmented tree objects to produce a stand-level AGB estimate.

## 2. Methods

### 2.1. Study Site and Data Collection

Our analysis is based on supervised segmentation of high-density point clouds from TLS and drone lidar collected within two days of each other under leaf-off conditions (16–18 April 2018) in a temperate mountain forest in the south Bohemia region of the Czech Republic (48°40′N 14°42′E; [18]). The site contains the 25 ha Žofín Forest Dynamics Plot (ZFDP). The ZFDP lies within the strictly protected (since 1838) forest national nature reserve and represents dense, structurally complex and variable old-growth forest dominated by European beech (*Fagus sylvatica*), with Norway spruce (*Picea abies*) and individuals of silver fir (*Abies alba*), and several other rare species [19]. At the ZFDP all free-standing woody plants with DBH > 1 cm have been mapped (X, Y, and Z coordinates of stem bases), measured (DBH) and identified to the species level using the ForestGEO protocol [19–21] in 2017 (see also Figure A2). The mean live stem density is >3000 individuals $\cdot$ ha$^{-1}$. TLS data were collected from 22 scanning positions within a 1 ha subplot in the ZFDP using a Lecia P20 ScanStation. The species composition and size distribution within the 1 ha subplot closely matched the composition of the 25 ha ZFDP (Figure 1). These data were co-registered using reflective targets, georeferenced and thinned using a 5 mm voxel grid. Mean absolute error on the target for the co-registration of single scans was 1 mm. Georeferencing of the TLS and field data is based on a permanent network of reference points with geodetically measured coordinates (total station). Ground and vegetation points in TLS data were classified in 3D Forest using the terrain-from-octree algorithm [15]. A further detailed description of TLS processing is in [15].

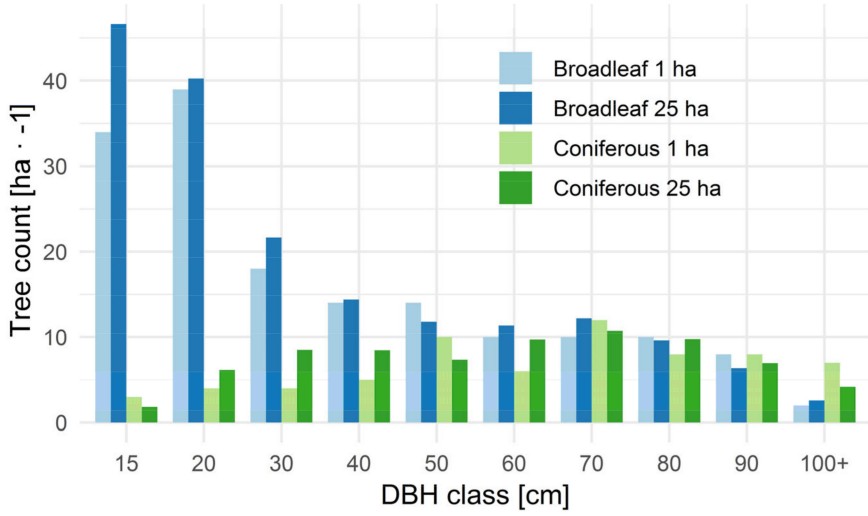

**Figure 1.** DBH distribution of trees on the 25 ha of ZFDP and on the 1 ha subplot covered by TLS.

### 2.2. Drone Lidar

We collected high-density drone lidar using the BPAR [5]. BPAR is a suite of sensors carried by a heavy-lift Aeroscout B100 helicopter drone. The airborne laser scanner was a RIEGL VUX-1 coupled to an Oxford Technical Solutions (OXTS) Survey +2 GPS and inertial motion unit (IMU). In the configuration used here, the GPS-IMU data stream was recorded at 250 Hz, and the instrument had access to the GPS and GLONASS constellations. During flight operations we collected an independent global navigation satellite system (GNSS) data stream on the ground using a Novel FlexPak 6 Triple Frequency + L-band GNSS receiver, and we used this data stream to differentially correct the OXTS GPS-IMU measurements in post-processing.

Nominal flight altitude was 110 m and the forward flight speed was 6 m $\cdot$ s$^{-1}$. The data were collected in six flights over two consecutive days during about 5 h of total flight time over a 1.72 km$^2$

area that included the ZFDP. There were 45 flight lines in the NE-SW direction and 45 flight lines in the NW-SE direction. We designed this flight plan to produce dense point coverage of stem and branch structure from a wide range of scan angles (−60° to 60°). Previous work has demonstrated that measurements of stem and branch structure require scan angles >30 degrees [5,7]. Measurement density in this point cloud is 4387 points · m$^{-2}$ within the ZFDP (Figure 2).

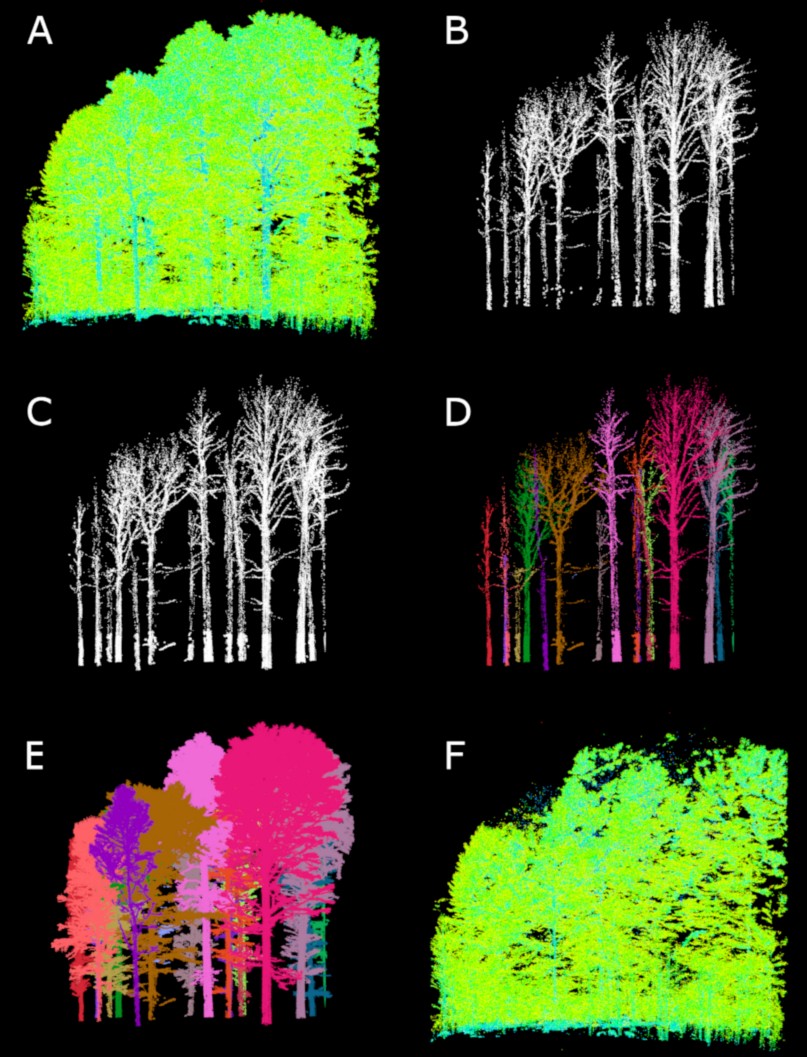

**Figure 2.** Lateral view of a 20 m × 20 m section of ultra-high-density drone lidar in a temperate mountain forest in the south Bohemia region of the Czech Republic: (**A**) The vegetation point cloud; (**B**) Skeleton points after low-intensity filtering; (**C**) Skeleton points + stem base points after spatial filtering and before segmentation; (**D**) Segmented skeletons I + II.; (**E**) Segmented trees after proximity point assignment; (**F**) Undifferentiated points.

### 2.3. Sources of Uncertainty in Drone-Lidar Point Clouds

Previous studies have demonstrated the utility of TLS for individual tree segmentation. Measurements from low-altitude drone flight differ from TLS data in ways that require additional processing before segmentation algorithms can be applied. The most important differences are point density, footprint size, and random error in point locations. Point densities from airborne applications decrease from the canopy top to the ground, because most laser energy is reflected by upper-canopy vegetation. Although measurement densities from drone lidar are enormous in comparison to traditional airborne laser scanning [22], they are about one order of magnitude less dense than typical TLS campaigns. Coupled with a decrease in sampling density from the canopy top

to the ground, measurement frequency at breast height is less than typically encountered by algorithms developed for TLS applications.

Footprint size in laser scanners is controlled by beam divergence and distance to the reflective target. The VUX-1 produces a 5 cm circular footprint at a distance of 100 m on a target that is perpendicular to the beam path. For the TLS instrument used here, the corresponding number is 2 cm. In practice the difference in footprint size between TLS and airborne applications is larger, because TLS measurements are reflected by objects closer to the sensor. For example, our nominal flight altitude was 110 m. This indicates that no returns were recorded <70 m from the scanner (given a 40 m maximum canopy height). This corresponds to footprint sizes in the 3.5–5.5 cm range at nadir. Beams emitted at wider scan angles can travel farther, and the corresponding footprint size is bigger. For example, a beam traveling along a 60 degree scan angle from 110 m altitude would be reflected by ground at a distance of 190.5 m from the scanner, at which the footprint size would be about 9.5 cm on a target perpendicular to the beam path. For TLS measurements in the 10–20 m range, footprint sizes are 2–4 mm. Thus, the difference in footprint size is about one order or magnitude.

Systematic and random error in the location of recorded laser returns is greater for airborne applications than terrestrial ones. This is because TLS data are acquired from a stationary position and the distance to reflective targets is shorter. Reported ranging accuracies for TLS instruments are < a few mm at distances of 100 m [23]. Fundamental range accuracy for the VUX-1 is 1 cm. In practice, the realized range accuracy within a given point cloud is probably in the 5–15 cm range. Multiple sources of uncertainty influence this value, including geolocation and altitude knowledge of the sensor at the time of laser firing, and dynamic offsets among flight lines. Footprint size and scan angle can introduce uncertainty into the location of discrete reflective surfaces within the beam path.

*2.4. Processing Workflow of the Drone-Lidar Point Cloud*

First, we separated the terrain points and vegetation points in FUSION/LDV software (Figure 3). Then we applied a return-intensity filter to the points classified as vegetation to distinguish reflections from larger stems and branches (hereafter called skeleton points) from partial reflections of small stems, branches and twigs that did not completely fill the laser footprint (hereafter called periphery points I, Figures 2B and 3). The intensity filter discarded all returns with 16-bit scaled reflectance intensity <55,000. This threshold was determined by visual inspection of the point cloud. We then searched within periphery points I to identify points that were ≤5 m aboveground and ≤5 cm from the nearest stem point. We assumed that these points were partial stem returns that were removed by intensity filtering and therefore labeled them as stem base points (Figures 2C and 3). Next, we applied the segmentation algorithm newly implemented in 3D Forest v. 0.5 [24] to the skeleton and stem base points in two iterations (Figures 2D and 3). In the first iteration the voxel size was set to 10 cm. The second iteration was applied to unsegmented points from the first iteration with a voxel size of 20 cm. Finally, we assigned unsegmented points (labeled rest of vegetation in Figure 3) to segmented tree objects iteratively (Figures 2E and 3) based on point proximity: when the nearest segmented tree was within a specified distance, the unlabeled point was assigned to that segmented tree. We repeated this process 20 times with the following settings, 10 times with a maximum distance of 10 cm, 5 times with a maximum distance of 15 cm and 5 times with a maximum distance of 20 cm.

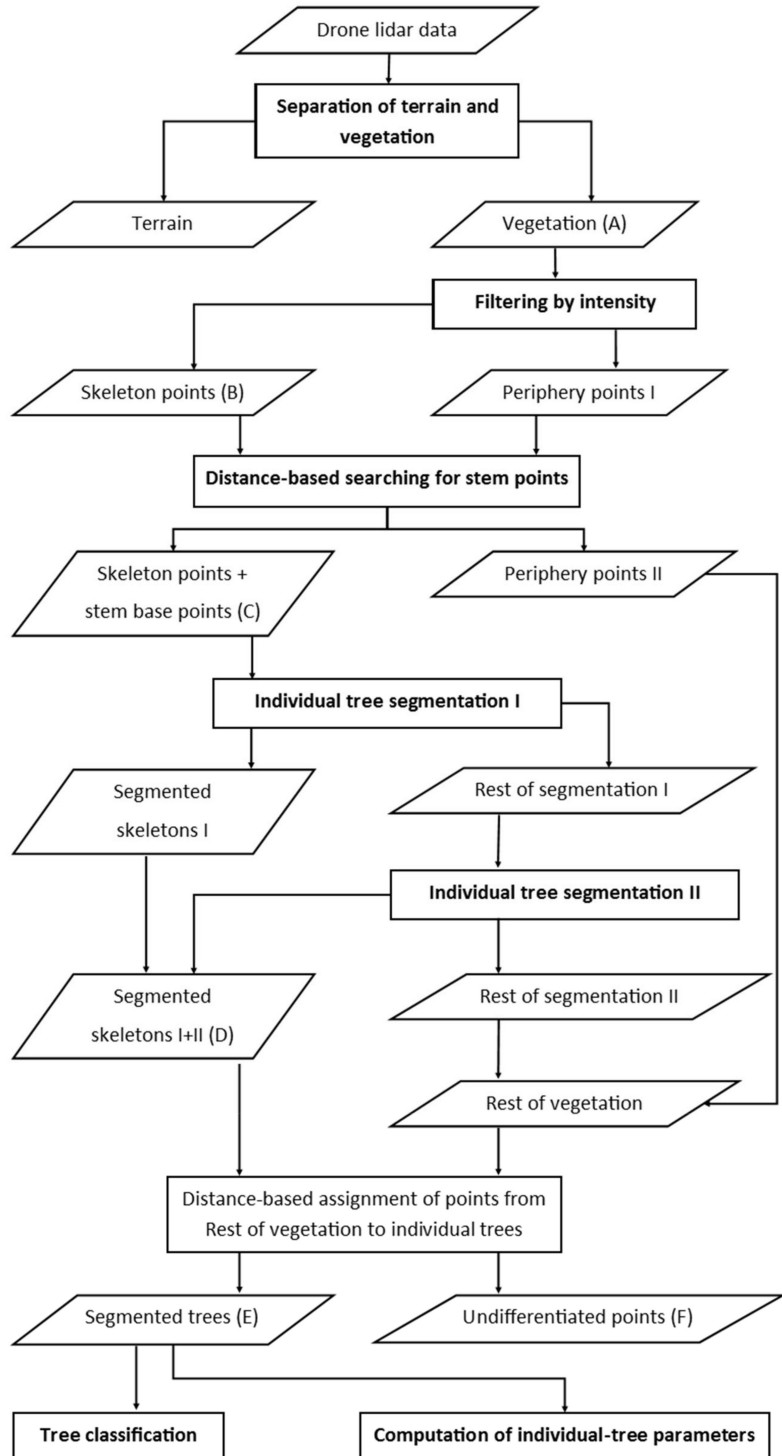

**Figure 3.** Workflow diagram for individual tree segmentation using ultra-high-density drone lidar. The capital letters in parentheses refer to the individual panels in Figure 2.

Based on the resulting segmentation (Figure 2E) we then computed the stem base position, tree height and DBH for each segmented tree object. Stem base position is derived using the median XY algorithm described in [15]. DBH is the median of 20 estimates obtained using 20 cm vertical slices of each tree object starting at 1.1 m aboveground in 5 cm increments (i.e., the first vertical slice was 1.1–1.3 m, the second was 1.15–1.35 m, etc.). The DBH estimate is based on a randomized Hough

transformation of the points within each 20 cm horizontal slice [15]. Tree height is the maximum Z coordinate among the points within the segmented tree object [15].

### 2.5. Random Forest Classification and AGB of Trees

We developed a Random Forest classifier to assign broadleaf or needleleaf labels to segmented tree objects [25]. We used four Random Forest parameters: the number of trees (n-estimators) was 1000, the criterion was Gini, the maximum depth and the maximum number of features were determined automatically. To train the Random Forest classifier, we used 125 randomly selected broadleaf trees and 125 randomly selected needleleaf trees. Because tree height varies by a factor of 2–3 in this forest, we expressed the height of each point relative to the maximum height within its associated tree object using,

$$z_{i,scaled} = \frac{z_{i,original} - z_{j,min}}{z_{j,max} - z_{j,min}} \tag{1}$$

where, $z_{i,scaled}$ is the scaled value for point $i$, $z_{j,min}$ and $z_{j,max}$ are the minimum and maximum height within segmented object $j$, and $z_{i,scaled}$ is the unscaled value for point $i$.

To quantitatively describe each segmented tree object, we divided every tree into 40 vertical layers and calculated the following three features for each layer: (i) diameter (cm), (ii) the standard deviation of Euclidean distances between points and the point-cloud mean for each layer in three dimensions, and (iii) the fraction of dense voxels. Each voxel was 10 cm on a side, and dense voxels were defined as those with >36 points. We used the diameter of each layer based on the assumption that broadleaf and coniferous species in this forest have different silhouettes when point clouds are projected along the sagittal plane. The second and third features were chosen based on the assumption that the distribution of returned laser energy will systematically vary between needleleaf and broadleaf trees in leaf-off condition (i.e., point density of needleleaf trees is greatest on the crown surface and decreases toward the crown center, and the opposite pattern occurs for broadleaf trees in leaf-off condition). We quantified aboveground biomass (AGB) for each segmented tree using the allometric scaling equations for European temperate forests in [26]. For all trees with the broadleaf classification we used the beech equation. For all trees with the needleleaf classification we used the spruce equation. For AGB computations we excluded segmented trees with a maximum height <10 m because visual examination indicates that many segmented objects <10 m in height were segmentation errors. We also excluded tree objects with DBH > 1.5 m. This value exceeds the size of the largest measured tree in the ZFDP. Segments of this size are therefore very likely to be commission errors (see example commission error in Figure 4, column four, row three). Therefore, our estimate of stand-level AGB is net of all errors in drone-lidar segmentations and classification of tree objects into needleleaf and broadleaf types.

### 2.6. Evaluating the Impact of Footprint Size and Noise on DBH Estimates

We quantified the impact of random error and footprint size on estimates of tree diameter using a data simulation. To quantify the impact of noise we generated circles over the range of 10–200 cm in diameter with 180 points that were regularly spaced along the circumference of each circle. We then randomized point locations by adding independent, normally distributed random error to each point in the X and Y dimensions with a mean equal to 0 and a standard deviation equal to 1, 2.5, 5 or 10 cm. We estimated the diameters of these circles using the randomized Hough transformation with 400 iterations, and compared the estimated diameters to the true values. We repeated this process 500 times for each diameter and measurement error scenario.

To quantify the impact of footprint size, we generated circles over the range of 10–200 cm in diameter using 3600 points regularly spaced along the circumference. We performed a convolution operation on these points using a binary filter whose length corresponds to a footprint size of 5, 10 or 15 cm (Appendix A, Figure A1). The filter proceeded along the circumference of each circle in increments of 2 cm, this was replicated for the four principal directions (which corresponds to the direction of airborne flight lines). At each increment, we estimated the distance measured by laser as

mean distance of all points within the extent of the 5, 10 or 15 cm simulated footprint and evaluated the effect of the difference between the estimated and real distance on the DBH estimation. Our analysis ignores some of the complexity of recorded footprints, such as variation in the across-beam laser intensity [27], and variation in footprint size as a function of scan angle on individual stems. As a point of reference, the VUX-1 produces a 5 cm footprint on an orthogonal surface at a distance of 100 m. At a scan angle of 60 degrees the laser beam path is likely to be longer and the expected footprint size is bigger. Most stem returns from high-density drone lidar are produced by wide scan angles [5,7]. The simulated footprint size range of 5–15 cm is therefore likely to encompass the range of footprint sizes on individual trees in our data.

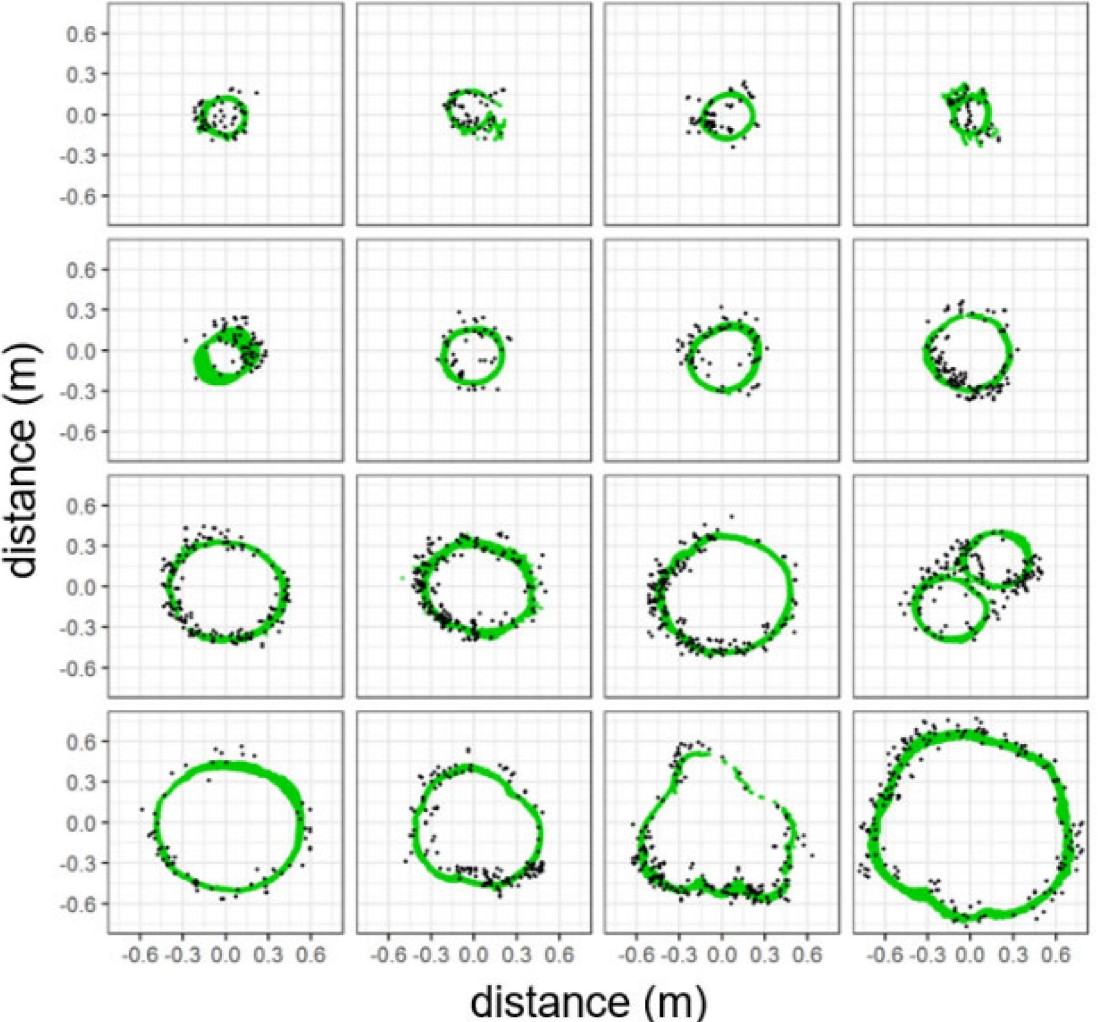

**Figure 4.** Representative comparisons between drone lidar (black points) and TLS (green points) within 20 cm vertical slices 1.1–1.3 m aboveground.

*2.7. Statistical Analysis*

First, we matched the field measurements and TLS segmented trees with trees segmented from drone lidar. The pairing was based on the Euclidean distance between segmented and known tree bases, and was visually examined for errors (Appendix A, Figure A2). We compared DBH and height from supervised segmentations of drone lidar to those obtained from TLS measurements and field data using ordinary linear regression. To determine whether there was a difference in the intercept or slope

of the relationship between broadleaf and needleleaf trees we used dummy-variables regression [28,29]. The regression model was:

$$Y_i = b_0 + b_1 X_1 + b_2 X_2 + b_3 X_1 X_2 + \varepsilon_i \tag{2}$$

where the response variable is DBH or height from drone-lidar segmentations, and $X_1$ is DBH or height from segmentations using TLS data, or DBH from field measurements. The variable $X_2$ is a binary indicator variable that takes the value of 0 if the segmented object was a needleleaf tree and a value of 1 if it was a broadleaf tree. The term $\varepsilon_i$ denotes a normally distributed random error. The labels used here are from field records, not the Random Forest classifier described above. Consider the relationship when $X_2 = 0$, which denotes needleleaf trees. When this occurs, $b_2 X_2$ and $b_3 X_1 X_2$ are 0, and the model simplifies to:

$$Y_i = b_0 + b_1 X_1 + \varepsilon_i \tag{3}$$

For broadleaf trees $X_2 = 1$, and the model is:

$$Y_i = (b_0 + b_2) + (b_1 + b_3)X_1 + \varepsilon_i \tag{4}$$

Thus, the term $b_2$ is a test of the hypothesis of no difference in the intercept between needleleaf and broadleaf trees, with the value of $b_2$ indicating the change in intercept for broadleaf trees. Similarly, the term $b_3$ tests the hypothesis of no difference in the slopes of the relationships, with the value of $b_3$ indicating the change in slope for broadleaf trees.

## 3. Results and Discussion

### 3.1. Individual Tree Segmentations from Drone Lidar and TLS

There is a linear relationship between the estimated DBH from segmentations applied to drone lidar and TLS data for the same individual trees (Figure 5). The intercept and slope of this relationship are 4.26 ($p = 0.575$) and 0.93 ($p < 0.001$) respectively ($r^2 = 0.981$, RSE = 8.91 cm, n = 85). Dummy-variables regression indicates that the slope and intercept are not statistically different between needleleaf and broadleaf trees ($p = 0.966$ for the change in intercept, and $p = 0.969$ for the change in slope). Our results are similar to those reported by [7] for 39 manually segmented trees in a temperate broadleaf forest in northeastern Europe, where the RSE was 4.24 cm and there was a 1.71 cm positive bias in comparison to TLS. Our analysis indicates that supervised segmentation of high-density measurements from low-altitude drone flight can produce estimates of individual tree diameter that are statistically indistinguishable from those acquired using TLS.

The relationship between height of individual trees segmented using drone lidar and TLS depends on needleleaf or broadleaf tree status ($p = 0.032$ for the change in intercept and $p < 0.001$ for the change in slope). The intercept and slope of the needleleaf height relationship are 6.72 ($p = 0.714$) and 0.58 ($p = 0.254$) respectively ($r^2 = 0.064$, RSE = 12.48 cm, n = 9). The lack of statistical significance in this relationship is influenced by the small number of needleleaf trees in our data and three outlying points apparent in the lower right of the height relationship in Figure 5. These trees have large estimated heights extracted from TLS segmentations, but smaller height values from drone-lidar segmentations. The intercept and slope for the broadleaf height relationship are −7.69 ($p < 0.001$) and 1.20 ($p < 0.001$), respectively ($r^2 = 0.907$, RSE = 2.35 cm, n = 76). The 95% CI on the intercept is −10.53–−4.85. The 95% CI on the slope is 1.10–1.29.

Our comparison of DBH and height from TLS and drone-lidar segmentations of individual trees indicates that diameter estimates are equivalent between these two data sources, but height estimates are not. The cause of the discrepancy in height estimates is preprocessing of the drone-lidar point cloud to remove low-intensity and partial returns (Figure 2B) and the process of determining which points are associated with tree segments. Low-intensity filtering removes points high in the canopy that are more likely to represent partial stem or branch returns. Removal of points by low-intensity filtering causes qualitatively different omission errors within broadleaf and needleleaf trees. For broadleaf trees,

this filter obscures the tops of understory crowns. For needleleaf trees, it reduces point density on the part of stem occluded by branches. This is the cause of the large underestimation errors among three needleleaf trees in the lower right of Figure 5.

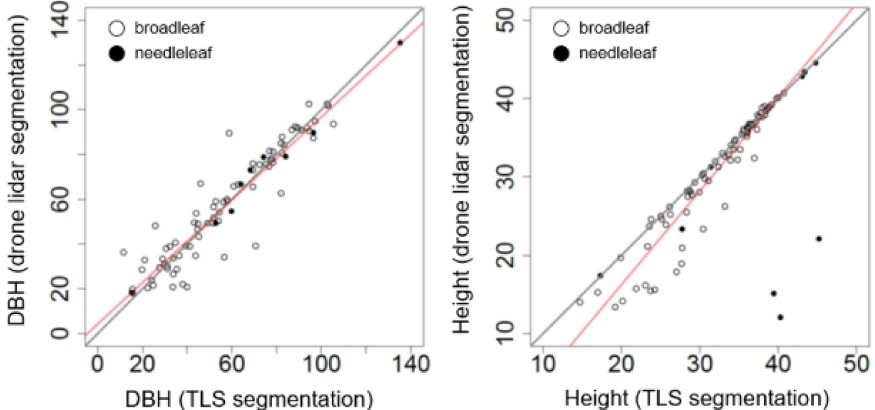

**Figure 5.** Relationships between DBH and height for trees segmented using high-density drone lidar and TLS. Left: The DBH relationship for individual trees segmented using drone lidar and TLS. Right: The height relationship for individual trees segmented using drone lidar and TLS. Height is derived from the segmented points that passed the intensity filter, not from all measurements acquired by drone lidar. Grey lines are the 1:1 relationship. Red lines are linear regression.

### 3.2. Sources of Error in DBH Estimates from Drone Lidar

We compared DBH from drone-lidar segmentations to field measurements of the same stems within the 25 ha ZFDP. The observed relationship is dependent on needleleaf versus broadleaf tree status (Figure 6). The intercept and slope of the needleleaf DBH relationship are 14.81 ($p < 0.001$) and 0.72 ($p < 0.001$) respectively ($r^2 = 0.945$, RSE = 16.93 cm, n = 470; Figure 6). The intercept and slope of the broadleaf relationship are 10.18 ($p = 0.001$) and 0.816 ($p < 0.001$) respectively ($r^2 = 0.949$ RSE = 13.7 cm, n = 1846; Figure 6). Slopes for the needleleaf and broadleaf relationships are significantly different from 1 (95% CI = 0.652–0.778 for needleleaf trees, and 0.791–0.841 for broadleaf trees).

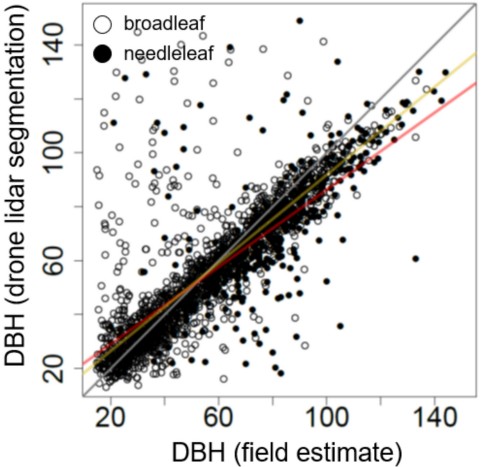

**Figure 6.** The DBH relationship for trees segmented using high-density drone lidar and field estimates. Grey line is the 1:1 relationship. Red line is the needleleaf-tree regression. Orange line is the broadleaf-tree regression.

For both broadleaf and needleleaf trees there is a clear tendency for overestimation of DBH among the smallest trees, and underestimation of DBH for the largest trees in comparison to field

measurements (Figure 6). The relationship between DBH from drone-lidar segmentations and field measurements predicts a 29.2 cm and 26.5 cm DBH for a needleleaf and a broadleaf tree with a 20 cm field estimate (a 46% and 32.5% overestimation error in comparison to field data, respectively). We assume that this relationship is not a statistical artifact, and that it is likely to be caused in part by random error in drone lidar data, because our simulations indicate that random error can positively bias diameter estimates of circles obtained using the randomized Hough transformation, especially for smaller diameters (Figure 7). When the standard deviation of the noise is 10 cm, the positive bias in diameter is considerable, decreasing exponentially from about 70% for 10 cm diameter circles to less than 5% for circles >40 cm in diameter (Figure 7). In fact, the difference between field estimates of DBH and DBH from drone-lidar segmentations exceeds the values expected for small trees under all noise scenarios considered. This indicates that other processes are contributing to the overestimation error in addition to random noise in point locations, or that the realized noise level in our post-processed lidar data is >15 cm.

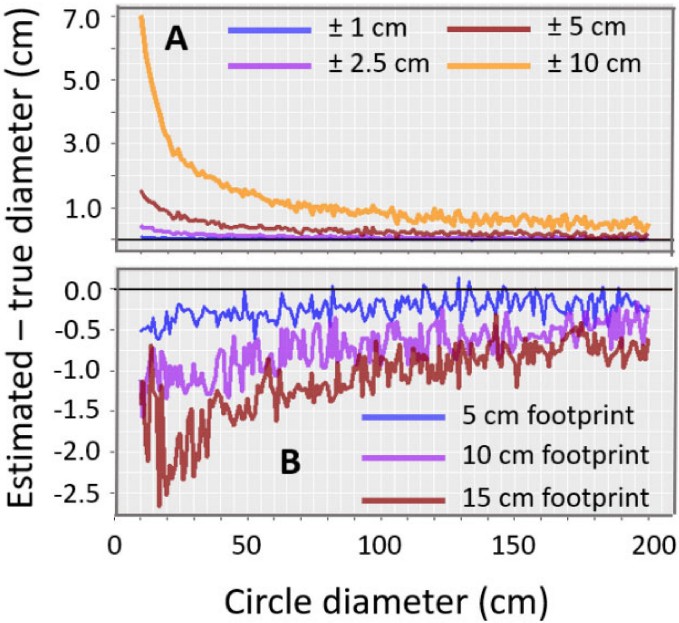

**Figure 7.** Simulated error in diameter estimates using the randomized Hough transformation. (**A**) Error in diameter due to random noise in simulated point locations under four noise levels (one sigma). (**B**) Error in diameter due to variation in footprint size.

For larger trees the DBH estimate is biased in the opposite direction. Needleleaf trees >52.9 cm DBH, and broadleaf trees >55.3 cm DBH experience an underestimation error in comparison to field measurements. The predicted underestimation error for a 100 cm DBH needleleaf tree is 13.2 cm. The corresponding broadleaf individual has an underestimation error of 8.2 cm. Our simulations indicate that in contrast to the positive bias in diameter estimates caused by random error in point locations, footprint size can cause an underestimation error of <1.5 cm for trees of this size when the footprint is close to 15 cm in diameter (Figure 7). The footprint-size bias approaches −1 cm as stem diameter increases. For the largest simulated footprints of 15 cm, the footprint-size bias is about 25% on a 10 cm diameter circle, and less than 4% for circles >40 cm in diameter (Figure 7). Thus, we cannot reject the possibility that variation in footprint size is contributing to underestimation error in DBH from drone-lidar segmentations of large trees, but it is clear that most of the underestimation error cannot be explained by variation in footprint size.

Another possibility is that field data are biased. This would explain why DBH estimates from segmentations using drone lidar and TLS agree, but drone-lidar segmentations are not in agreement with field data (and by inference, neither are our DBH estimates from TLS). The field estimate of DBH

is itself based on a model. This is because the measurement requires placing a tape around the tree stem to measure the convex hull, or using calipers to measure the maximum distance across the tree stem. Diameter tapes convert the length of the convex hull into a stem diameter under the assumption that the stem is a perfect circle. Because larger trees are more likely to depart from this assumption than smaller trees (see Figure 4), and departures from the assumption of a perfect circle result in positive bias to diameter estimates, errors in field data contribute to the uncertainty in the DBH relationship between drone-lidar segmentations and field measurements. Determining whether departure from idealized form is a cause of the discrepancy between DBH estimates from drone-lidar segmentations and TLS data is a research priority.

### 3.3. Ability of Drone Lidar to Produce a Stand-Level Tree Inventory

There were 4292 trees in the 25-ha plot with DBH > 15 cm. There were 2237 (51%) of these trees that were segmented using drone lidar and 2175 trees that were not (49%; Table 1). Most (58.5%) omitted trees were <40 cm DBH (Table 1). In larger DBH classes the omission rate was less. For example, in DBH classes >50 cm the omission rate was 9.5–16.1%, depending on DBH class, and automated segmentation correctly identified 86% of trees >50 cm DBH (Table 1). These numbers reflect our supervised segmentation without manual intervention.

**Table 1.** Number of trees in ten DBH classes in the 25 ha ZFDP and the corresponding number of trees segmented within each DBH class.

| DBH Class | 15–20 | 20–30 | 30–40 | 40–50 | 50–60 | 60–70 | 70–80 | 80–90 | 90–100 | >100 | Total |
|---|---|---|---|---|---|---|---|---|---|---|---|
| No. field | 1157 | 886 | 536 | 355 | 286 | 306 | 295 | 272 | 170 | 149 | 4412 |
| No. segmented | 78 | 281 | 332 | 268 | 240 | 267 | 267 | 234 | 144 | 126 | 2237 |
| % segmented | 6.74% | 31.72% | 61.94% | 75.49% | 83.92% | 87.25% | 90.51% | 86.03% | 84.71% | 84.56% | 50.70% |

### 3.4. Aboveground Biomass Estimates from Individual Tree Segmentations

We quantified how size-dependent tree recognition rates and DBH estimates impact individual-based AGB estimates. We used a Random Forest classifier to label segmented tree objects as broadleaf or needleleaf trees. The overall accuracy of the Random Forest classifier was 85.9% (Table 2). The largest error was an error of commission for needleleaf trees (36.9%). Because allometric scaling equations of broadleaf trees in this forest predict more AGB than needleleaf trees of the same size [26], needleleaf trees that are incorrectly identified as broadleaf trees cause a positive AGB error with respect to field data, and vice versa.

**Table 2.** Classification accuracy for the Random Forest algorithm applied to individual tree segmentations using high-density drone lidar.

| | | Field Reference | | | | |
| | | Broadleaf | Needleleaf | Total | Omission Error | Commission Error |
|---|---|---|---|---|---|---|
| Classification | Broadleaf | 1526 | 52 | 1578 | 15.2% | 3.3% |
| | Needleleaf | 273 | 465 | 738 | 10.1% | 36.9% |
| | Total | 1799 | 517 | | Overall accuracy = 85.9% | |

All errors in segmentation, tree classification, and DBH estimation contribute to uncertainty in stand-level AGB in our analysis. For example, omitted trees cause some individuals to be excluded from stand-level estimates. Misclassifications cause the wrong allometric scaling equation to be applied to the segmented tree, and DBH estimation errors will positively bias AGB for small trees and negatively bias AGB for large trees. Because large trees are disproportionately important contributors to stand-level AGB [30], the negative DBH error for large trees will reduce expected AGB in comparison to field data. Integrating across all sources of error, segmentations using drone lidar contain 76% of the field-estimated AGB in the 25 ha plot (Figure 8). This is larger than the percentage of correctly segmented trees (51%) because larger trees are more likely to be correctly segmented (Table 1), and because DBH

estimates for smaller trees that are more likely to be omitted are positively biased. Because we know which trees were omitted by segmentations, we can compute the AGB of the omitted trees to determine what fraction of the missing 24% of AGB is exclusively due to errors of omission. This number is 17.6%. The remaining 5.6% is due to DBH estimation error, misclassification of segmented tree objects into broadleaf and needleleaf categories and other sources of uncertainty.

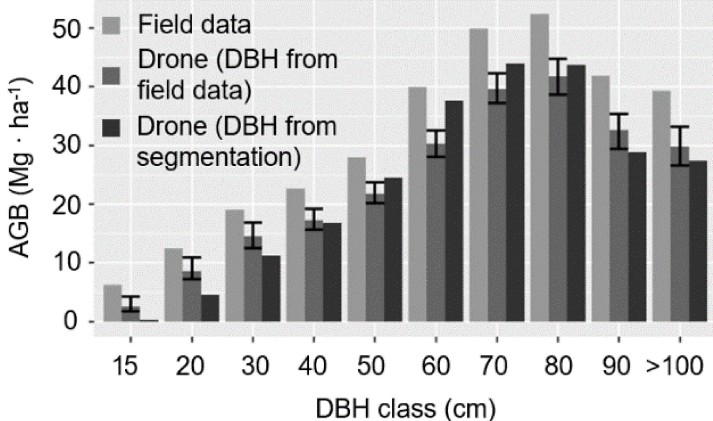

**Figure 8.** AGB in ten diameter size classes. For each size class, the three bars compare AGB from all trees mapped in the field (left) to the subset of trees segmented using drone lidar with AGB computed using field estimates of DBH (center), and the subset of trees segmented using drone lidar with AGB computed using DBH estimates from drone lidar segmentations (right). The difference between the left and center bars is due exclusively to errors of omission. The difference between the center and right bars is due exclusively to errors in DBH estimation.

## 4. Conclusions and Recommendations

The largest source of uncertainty in our estimate of stand-level AGB is omission errors in tree segmentation. Using supervised segmentation applied to high-density drone lidar, we detected 51% of the individuals and 76% of the AGB within individuals >15 cm DBH. This number is net of all segmentation and Random Forest classification errors necessary to apply allometric scaling equations to segmented trees. We showed that 73.3% of the bias (17.6 percentage points) was due to omission errors, and the remaining 6.4 percentage points were due to DBH estimation errors, variation in footprint size and other sources of uncertainty. Visual examination of high-density point clouds demonstrated that most omitted trees were present with sufficient point densities for automated segmentation. These trees are being removed by intensity filtering prior to deploying the segmentation algorithm. Changes to airborne data collection and pre-processing could therefore reduce errors of omission. For example, reducing the flight altitude and increasing the number of flight lines will reduce laser ranges, footprint size and random error in point locations. It will also increase the mean intensity of recorded laser returns. More sophisticated filtering of points using geometrical and intensity characteristics [31] and other information, including scan angle and the number of returns per emitted laser pulse, will retain more points for automated segmentation.

Our analysis demonstrates that high-density measurements from low-altitude drone flight can produce information about segmented individual trees that is in some aspects comparable to TLS, despite much lower point densities, attenuation of laser energy within the canopy volume, larger errors in point locations, and larger footprint size. These data can be collected rapidly throughout areas large enough to produce landscape-scale estimates that could augment or replace manual field inventories, and become suitable for calibration and validation of current and forthcoming space missions [11,12]. Our analysis does not address whether large-area estimates of AGB from individual tree segmentation are more precise or less biased than area-based extrapolations using traditional methods, and does not address the propagation of sources of uncertainty through area-based summaries [32–34].

**Author Contributions:** M.K. processed the drone lidar and TLS data, performed the simulation analysis and wrote the first draft of the manuscript. K.K., K.C. and J.R.K. contributed to writing the manuscript. M.K., K.K., and K.C. collected the TLS data. A.M. performed the tree classification and accuracy evaluation. J.R.K. oversaw drone lidar data collection and processing, and performed the statistical analysis. All authors have read and agreed to the published version of the manuscript.

**Funding:** This research was funded by Inter-Action project LTAUSA18200, by the Institute at Brown for Environment and Society at Brown University and by funds provided to Brown University by Peggy and Henry D. Sharpe Jr.

**Acknowledgments:** We gratefully acknowledge the ForestGEO Annual Analytical Workshop program funded by the Smithsonian Institution. We thank Christoph Eck, Christoph Falleger, Benedikt Imbach, and Carlo Zgraggen.

**Conflicts of Interest:** The authors declare no conflict of interest.

**Appendix A**

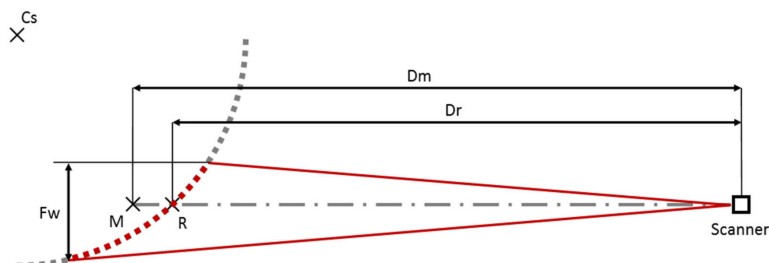

**Figure A1.** Top view sketch of the simulation of footprint size impact to the DBH estimate. *Cs* is center of stem, grey points represent the stem surface, red points represent the stem surface covered by a single laser beam. Red lines show laser beam divergence, *Fw* is laser footprint width. *Dr.* is real distance from the scanner to the center of laser beam on the stem surface. *Dm* is measured distance from the scanner, computed as the mean distance of all red points to the scanner. *M* is the location of simulated point. *R* is the location of center of the laser beam footprint.

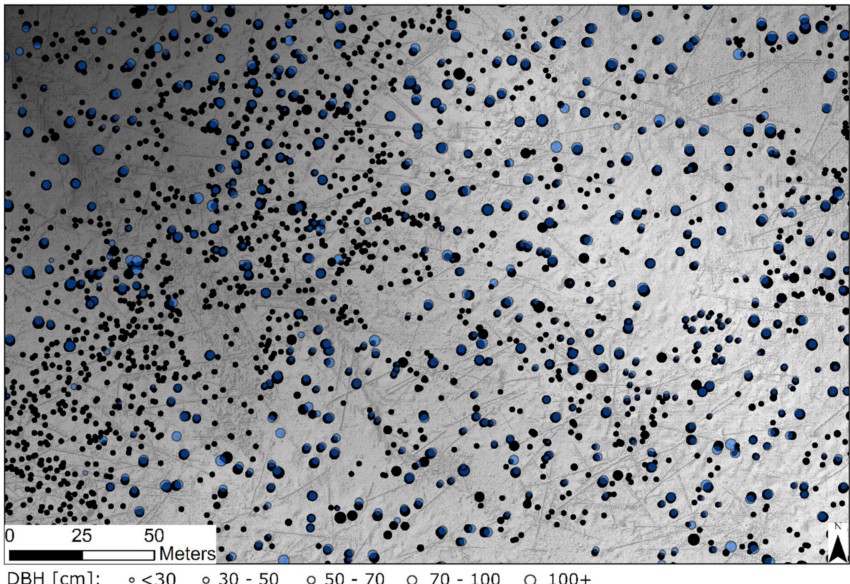

**Figure A2.** Small section of the ZFDP stem position map demonstrating correspondence between locations of segmented trees and field-mapped stems. Black points are stem positions from field measurements. Blue points are stem positions from a supervised segmentation of high-density drone lidar. Note: Small positional mismatch is caused by uncertainty in field measurements. The background is a high-resolution digital terrain model from drone lidar.

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
