# Peer review of "Supervised Segmentation of Ultra-High-Density Drone Lidar for Large-Area Mapping of Individual Trees"

_remotesensing, doi:10.3390/rs12193260_

Round 1
Reviewer 1 Report
Individual tree segmentation and extraction is a tough task in drone lidar point cloud data analysis. This study used the 3D Forest segmentation algorithm to retrieve individual trees, and the accuracy for the individual tree segmentation is evaluated by comparing with TLS estimates and field measurement. The findings will interest the remote sensing researchers and forest inventory department. The following details should be revised.
(1) The figures in the body text should be close to their first citation and arranged sequentially.
(2)Lines 163, "The intensity filter discarded all returns with 16-bit scaled 162 reflectance intensity < 55,000 ". The reflectance intensity could be weakened due to the larger scanning angles even if the tree stem size is same. Was any correction made to the intensity difference or just used a fixed value of reflectance intensity for the filtering operation?
(3) In section 2.4, I just wonder how the segmentation results were sensitive to and impacted by several threshold values used in this study?
(4) It is suggested that the features used in the random forest classification is listed in a table, and the preparation of the training samples should be addressed more in detail.
(5)Is the segmentation the best one for drone lidar points? If it is possible, a comparative study may be more persuadable.
Reviewer 2 Report
I have no special comments on the work. I think that it was done methodologically correctly and the results were presented in a clear way.
Author Response
There were no comment of the reviewer needed response.
Reviewer 3 Report
I would like to congratulate the authors for their excellent work, this study has its eminent importance in the forest context of collecting inventory data and its improvement will serve as a basis for future works with more consistent results.
Author Response

(The authors gave the same response as above.)

Reviewer 4 Report
The manuscript deals with an important and interesting topic of using Drone Lidar data for forestry applications. The manuscript need some improvements in the method and discussion chapter - see detailed comments below.
Finally, an additional language editing could improve the readability of the manuscript.
Abstract:
line 13-14: to ultra-high-density drone lidar in a -> to ultra-high-density drone lidar data in a
Methods:
line 91: in which way the segmentation is supervised?
line 104: the description of the available field data is missing
line 153: you assume a range accuracy for the VUX1 of 5-15cm. Do you have any facts/data that support this assumption?
line 164: it is not clear how you derived the stem base points. Especially the condition ≤ 5 cm from a stem point. What does this mean? Is it the 2D distance or the shortest distance ?
line 180: if the tree height is defined as the max Z value within all segmented tree points, it must be assumed that the segmentation is correct. Holds this assumption for overgrown trees?
chapter 2.5. Random Forest Classification and AGB of Trees: the used parameter are only described roughly. Please describe the used parameter in a re-producable way.
chapter 3.1. Individual Tree Segmentations from Drone Lidar and TLS
Concerning the comparison of tree heights from TLS and drone liDAR I suggest to discuss also the fact that in TLS data you have several occlusions in the top canopy regions...
The discussion chapter 3.1 and 3.2 don't include any connex to the literature/other work, which should be changed.
Reviewer 5 Report
This paper studied the rapid large-area mapping of individual trees using drone LiDAR and random forest classification. The errors are analyzed for different kinds of trees.
- Please explain why the “Random Forest classifier” is used to label segmented trees. Did you try other classifiers?
- “The challenge is the consistent application of segmentation algorithms to large data sets.” Why the size of the dataset is a challenge? The test on the larger dataset is the contribution of the paper, but it is weak. How large is the dataset can be called “large”?
- Some citations have incorrect formats.
